# Fabrication of Gelatin-ZnO Nanofibers for Antibacterial Applications

**DOI:** 10.3390/ma14010103

**Published:** 2020-12-29

**Authors:** Nataliya Babayevska, Łucja Przysiecka, Grzegorz Nowaczyk, Marcin Jarek, Martin Järvekülg, Triin Kangur, Ewa Janiszewska, Stefan Jurga, Igor Iatsunskyi

**Affiliations:** 1NanoBioMedical Centre, Adam Mickiewicz University, Wszechnicy Piastowskiej 3, 61-614 Poznań, Poland; lucja.przysiecka@amu.edu.pl (Ł.P.); nowag@amu.edu.pl (G.N.); marcin.j@amu.edu.pl (M.J.); stjurga@amu.edu.pl (S.J.); 2Institute of Physics, University of Tartu, W. Ostwaldi Str 1, 50411 Tartu, Estonia; martin.jarvekylg@ut.ee (M.J.); triin.kangur@ut.ee (T.K.); 3Faculty of Chemistry, Adam Mickiewicz University in Poznan, Uniwersytetu Poznanskiego 8, 61-614 Poznan, Poland; eszym@amu.edu.pl

**Keywords:** ZnO NPs, gelatin nanofiber, antibacterial activity, morphology, luminescence

## Abstract

In this study, GNF@ZnO composites (gelatin nanofibers (GNF) with zinc oxide (ZnO) nanoparticles (NPs)) as a novel antibacterial agent were obtained using a wet chemistry approach. The physicochemical characterization of ZnO nanoparticles (NPs) and GNF@ZnO composites, as well as the evaluation of their antibacterial activity toward Gram-positive (*Staphyloccocus aureus* and *Bacillus pumilus*) and Gram-negative (*Escherichia coli* and *Pseudomonas fluorescens*) bacteria were performed. ZnO NPs were synthesized using a facile sol-gel approach. Gelatin nanofibers (GNF) were obtained by an electrospinning technique. GNF@ZnO composites were obtained by adding previously produced GNF into a Zn^2+^ methanol solution during ZnO NPs synthesis. Crystal structure, phase, and elemental compositions, morphology, as well as photoluminescent properties of pristine ZnO NPs, pristine GNF, and GNF@ZnO composites were characterized using powder X-ray diffraction (XRD), FTIR analysis, transmission and scanning electron microscopies (TEM/SEM), and photoluminescence spectroscopy. SEM, EDX, as well as FTIR analyses, confirmed the adsorption of ZnO NPs on the GNF surface. The pristine ZnO NPs were highly crystalline and monodispersed with a size of approximately 7 nm and had a high surface area (83 m^2^/g). The thickness of the pristine gelatin nanofiber was around 1 µm. The antibacterial properties of GNF@ZnO composites were investigated by a disk diffusion assay on agar plates. Results show that both pristine ZnO NPs and their GNF-based composites have the strongest antibacterial properties against *Pseudomonas fluorescence* and *Staphylococcus aureus*, with the zone of inhibition above 10 mm. Right behind them is *Escherichia coli* with slightly less inhibition of bacterial growth. These properties of GNF@ZnO composites suggest their suitability for a range of antimicrobial uses, such as in the food industry or in biomedical applications.

## 1. Introduction

Among semiconductor materials, zinc oxide (ZnO) is a well-known wide-bandgap semiconductor with a bandgap energy (3.37 eV at room temperature) and a large exciton binding energy (60 meV) [1]. The excellent properties of ZnO—mainly optical, luminescent, catalytic, and electrical—allow its use in thin film transistors, photodetectors, light-emitting diodes, solar cells, sensors, etc. [2,3]. Moreover, due to its good biocompatibility, low cytotoxicity, high surface to volume ratio with enhanced surface reactivity, and antistatic, antimicrobial, antibacterial, and antifungal properties, ZnO found broad application in biomedicine as a drug carrier, a biomarker for cell labelling, a biosensor, and an antibacterial agent [4,5,6]. The material’s functional properties can be achieved by modification of their crystal structure, e.g., by incorporation of impurities into the crystal structure or by surface modification/construction of the materials with the core-shell structure, as well as by obtaining organic/inorganic or inorganic/inorganic composites [7,8,9,10,11].

Polymeric materials are good carriers for incorporating or covering the functional agents (nanofillers) for achieving new physicochemical properties [12]. They can enhance the optical and mechanical properties, biocompatibility, antibacterial activity, or water stability of the resulting materials [13]. For example, Beek et al. obtained composites based on ZnO nanoparticles and nanorods with conjugated polymer (MDMO-PPV) for potential application in solar cells [14].

Several polymeric materials, such as collagen and gelatin, have been investigated in the past due to their low toxicity, excellent film forming ability, abundancy, high biocompatibility, and biodegradability in physiological environments [15,16]. Gelatin is one of the most studied biopolymers, and it is widely used in food (mainly as biodegradable packaging), cosmetics, pharmaceutical, and medical applications. Gelatin scaffolds consisting of randomly oriented fibers are an ideal candidate for skeletal muscle tissue engineering, developing biodegradable food packaging, and substituting for petroleum-based polymers [17,18]. Gelatin can be obtained and used in many forms, such as films, micro- or nanoparticles, and fibers [17,19].

Despite the above-mentioned benefits, gelatin has some significant disadvantages, such as its poor water-barrier and mechanical properties, which limits its application, for instance, in food packaging. To overcome these difficulties, alternative solutions have been developed, e.g., its incorporation into different kinds of nanoparticles to introduce a synergistic effect of both components. Nowadays, the use of the gelatin matrix as an organic additive in composites with inorganic nanoparticles is of great interest for bioapplication, mainly as antibacterial agents [16,18,19]. Nourbakhsh et al. obtained composites based on gelatin/ZnO nanocomposite films and studied their antibacterial properties [20]. The authors showed that nanocomposite films containing 0.5% of ZnO nanoparticles had the most resistance against *Staphylococcus aureus* and *Escherichia coli* bacteria.

The antibacterial activity depends on many factors, especially morphological (particle size, shape, and concentration). The small size and high surface area of nanoparticles help to achieve higher adsorption of the parcels on the cell membrane and more effective penetration of particles into the bacteria. Chemical synthesis approaches allow nanoparticles (including ZnO) with good characteristics (high crystallinity, dispersion, stability) to be obtained [21].

In this research, we obtained composites based on gelatin nanofibres and ZnO nanoparticles. First of all, we describe a simple synthetic route for the preparation of composites, consisting of gelatin nanofibres obtained by an electrospinning technique and ZnO nanoparticles (NPs) prepared by a facile sol-gel approach. The composite’s crystallinity, morphology, textural and luminescent properties, as well as its antibacterial activity toward Gram-positive (*S. aureus, B. pumilus*) and Gram-negative (*E. coli*, *P. fluorescens*) bacteria were assessed.

## 2. Materials and Methods

### 2.1. Materials

Zinc acetate dihydrate (Zn(CH_3_COO)_2_ × 2H_2_O, Sigma-Aldrich, Poznan, Poland); sodium hydroxide (NaOH, Stanlab, Lublin, Poland); methanol (Avantor, Gliwice, Poland); ethanol (Avantor); and gelatin type A from porcine skin, _D_-(+)-glucose, and glacial acetic acid (CH_3_COOH) (Labochema Estonia OÜ, Tartu, Estonia), were used as starting materials for the synthesis of ZnO NPs, gelatin nanofiberse, and GNF@ZnO composites (gelatin nanofibers (GNF) with zinc oxide (ZnO) nanoparticles (NPs)).

### 2.2. Preparation of Gelatin Fibers

Gelatin nanofibers (GNF) were prepared by electrospinning, following the method described by Siimon et al. [22]. Glucose and type A gelatin from porcine skin were used to prepare the solution for electrospinning. Glucose was mixed with gelatin in a ratio of 1:10. The mixture was dissolved in 10 M acetic acid at room temperature (RT) during vigorous stirring on a magnetic stirrer (Labochema Estonia OÜ, Tartu, Estonia). Fibrous scaffolds were prepared by electrospinning under the following condition: a 5 mL syringe containing gelatin solution was pumped at speeds of 5–7 μL/min at 17.5 kV. Aluminum foil was used as the grounded target, placed 14.5 cm away from the syringe needle tip. Fibrous gelatin scaffolds were removed from the foil and stored in Petri dishes (Labochema Estonia OÜ, Tartu, Estonia) until thermal treatment. Subsequent cross-linking was carried out by placing the fibrous scaffolds in an oven for 3 h at 175 °C to avoid gelatin degradation and to ensure proper cross-linking.

### 2.3. GNF@ZnO Composite Synthesis

GNF@ZnO composites were obtained by synthesis of ZnO nanoparticles (ZnO NPs) on previously prepared gelatin nanofibers pieces (~2.25 cm^2^). ZnO NPs were prepared using an adapted sol-gel procedure based on the work of W. J. E. Beek et al. [15]. The procedure used for the preparation of GNF@ZnO was as follows: 13.4 mmol of zinc acetate dihydrate (Zn(CH_3_COO)_2_ × 2H_2_O)) was dissolved in 125 mL methanol (MeOH) at constant temperature (60 °C) with vigorous stirring. After dissolving of the zinc acetate dihydrate, the solution of NaOH (23 mmol) in 60 mL methanol was added. Then, the gelatin nanofibers pieces were added to the reaction mixture. The formation of GNF@ZnO composite was stopped after 2, 5, and 24 h of standing GNF in the mixture. The obtained GNF@ZnO composites were separated and washed twice with methanol and dried for 12 h at 50 °C before further characterization.

### 2.4. Characterization

The detailed morphological characterization of ZnO NPs was carried out by means of high-resolution transmission electron microscopy (HRTEM; JEOL ARM 200F, JEOL, Tokyo, Japan). Specimens were prepared by dispersing ZnO NPs in ethanol under ultrasonic stirring, then dropping the solution onto a copper grid and evaporating the solvent naturally in the air. Furthermore, the morphology of GNF and GNF@ZnO composites and their chemical composition were also studied by scanning electron microscopy (SEM, JEOL, JSM-7001F, JEOL, Tokyo, Japan) equipped with an energy dispersive X-ray (EDX) analyzer (Oxford Instruments XMax 80 mm^2^ detector). Powder X-ray diffraction (XRD) studies of the ZnO NPs were carried out on an Empyrean (PANalytical, Malvern, UK) diffractometer using Cu Kα radiation (λ = 1.54 Å), a reflection-transmission spinner (sample stage), and a PIXcel 3D detector, operating in the Bragg–Brentano geometry. Scans were recorded at RT (300 K) in angles ranging from 20 to 80° (2Theta) with a step size of 0.006 and continuous scan mode. The photoluminescence (PL) data were recorded using a HeCd laser (325 nm) and a USB Ocean Optic spectrometer (Kimmon Koha, Tokyo, Japan). Fourier transform infrared spectroscopy (FTIR) spectra were measured in the 400–4000 cm^−1^ range with 2 cm^−1^ resolution using a Bruker VERTEX 70 spectrometer (Bruker Baltic OÜ, Tallinn, Estonia) with an attenuated total reflection (ATR) accessory. The N_2_ adsorption/desorption isotherms were measured at 77 K on a Quantachrome Nova 1000 apparatus (Anton Paar, Warsaw, Poland). The specific surface area of pristine ZnO NPs was determined using the BET (Brunauer–Emmett–Teller) method. The total volume of pores (at p/p0 = 0.98) was calculated using the single point model. The average pore radius was determined by applying the Barrett–Joyner–Halenda (BJH) method to the desorption branch of the isotherm. Thermal analysis (TG/DTA) was performed using a SETARAM SETSYS 12 thermogravimetric analyzer (Comef, Katowice, Poland). The analysis was carried out in a temperature range of 20–1000 °C, in an air atmosphere, and at a constant heating rate of 5 °C/min.

### 2.5. Antibacterial Activity of the Obtained GNF@ZnO Composites

To evaluate the antimicrobial activity of the tested materials against *Escherichia coli*, *Staphylococcus aureus*, *Bacillus pumilus*, and *Pseudomonas fluorescens*, the agar disk diffusion method was applied [23,24]. In brief, 1 mL of fresh culture having 10^6^ CFU/mL was added to 9 mL of freshly prepared LB (Luria-Bertani) medium and incubated overnight on a rotary shaker (37 °C, 230 rpm). To prepare the Petri plates, 20 mL of melted LB broth and Lennox agar medium (Sigma Aldrich, Darmstadt, Germany) was poured into sterile plates and cooled. Next, a disk made of GNF@ZnO composites (GNF with ZnO NPs after 2, 5, and 24 h), pristine gelatin nanofibers, and 1 mg of pristine ZnO NPs, respectively, were placed on agar plates inoculated with bacterial culture. The zones of inhibition growth were measured after 24 h of incubation (37 °C). All experiments were done in triplicate.

## 3. Results

### 3.1. Characterization of ZnO NPs

#### 3.1.1. TEM and XRD Analyses

The morphology of pristine ZnO NPs was studied by high-resolution transmission electron microscopy (HRTEM). The majority of obtained ZnO NPs were almost perfectly spherical and uniform in their size distribution, with a mean value of nanoparticles’ diameter of about 7 nm (Figure 1a).

The phase purity and composition of ZnO NPs obtained by a sol-gel method was examined by the XRD technique. Figure 1b presents typical XRD data of ZnO NPs. All diffraction peaks were indexed to the hexagonal phase of ZnO with main (100), (002), (101), (102), (110), (103), (112), and weak (202) crystal planes (corresponding to the crystallographic data in the ICDD PDF-4+ 2019 database). No characteristic peaks of impurity phases except ZnO were found, which revealed the good crystalline nature of the samples. The broadening of the peaks can be attributed to the small particle size of the synthesized ZnO NPs [25].

#### 3.1.2. Textural Properties

It is known that due to their small size, nanoparticles have a relatively larger surface area than the bulk phases [26]. This feature predestines the use of nanoparticles in many scientific fields, including biological technologies [27]. High surface area leads to effective adsorption of biomolecules, which is very important for biosensors and drug delivery, and for enhancing antibacterial efficiency. Thus, the calculation of surface characteristics is important for the study of antibacterial activity.

To study the textural properties, the N_2_ adsorption–desorption analysis was carried out on dried ZnO NPs. Results show that the used procedure of ZnO NPs synthesis allowed material with a surface area of 83 m^2^/g, a pore volume of 0.17 cm^3^/g, and an average pore diameter of 8.3 nm to be obtained. The obtained ZnO NPs exhibited isotherms of Type IV, according to IUPAC classification, which is typical for mesoporous materials (Figure 2a). The material showed a narrow pore size distribution in a range from 3 to 9 nm.

### 3.2. Characterization of GNF@ZnO Composites

#### 3.2.1. Morphology of the Gelatin Nanofibers and GNF@ZnO Composites

The morphology of the pristine gelatin nanofibers and GNF@ZnO composites was studied by scanning electron microscopy and is presented in Figure 3. The fibers obtained by the electrospinning technique were multifiber ones. The fibers were entangled and they were arranged randomly. The diameter of one fiber was approximately 1 µm (Figure 3a). Generally, to obtain layered composites with good quality characteristics (i.e., homogeneous layer on a substrate), methods based on physical deposition are used. Sputtering or physical vapor deposition allows thin homogeneous layers to be obtained; however, these approaches are technically advanced and costly. Materials obtained by deposition techniques find application mainly in solar cells, diodes, lasers, etc. [28]. In our case, the fabrication of GNF@ZnO composites was performed by a simple approach based on maintaining the gelatin nanofibers in a Zn^2+^ methanol solution for 2, 5, and 24 h. After maintaining the gelatin nanofibers for 2 h (as an example) it was clearly seen that ZnO NPs homogeneously covered over the whole nanofibers surface (Figure 3b, inset). The SEM images for 5 and 24 h are presented in Appendix A.

#### 3.2.2. Elemental Composition of Studied GNF@ZnO Composites

The elemental identification of GNF@ZnO composites obtained at different maintaining times was determined by EDX spectroscopy. The quantitative results were calculated from three independent regions of the samples and the average values are summarized in Table 1. The detailed elemental analysis showed that the amount of Zn increased together with the increase of maintaining time. Thus, by regulating the synthesis time we could control the amount of ZnO NPs attached to gelatin nanofibers surfaces.

#### 3.2.3. Thermal Analysis of GNF@ZnO Composites

The results of the thermogravimetric analysis of GNF@ZnO composites are presented in Appendix A and Table 2. The resulting TG/DTA thermograms were used for determination of special temperatures of composite weight loss and maximum weight loss of the composites caused by thermal decomposition. The thermal analysis of GNF@ZnO composites in air indicated three weight losses on TG profiles (Appendix A). The first one (below 100 °C), accompanied by the endothermic effect on DTA curve, was a result of the evaporation of moisture. GNF@ZnO 2 h and GNF@ZnO 5 h composites indicated similar content of water (~7 wt.%), whereas the GNF@ZnO 24 h was more hydrophobic. The next two weight losses, in the range 200–550 °C, corresponded with the decomposition of gelatin nanofibers. The decomposition of gelatin nanofibers was presented by the exothermic peaks due to the oxidation of nanofibers, followed by the endothermic peaks attributed to the desorption of the oxidation products. The first stage of degradation was due to degradation of gelatin nanofibers not covered by ZnO NPs (the peaks at 300 °C and 360 °C) and the second one is attributed to decomposition of nanofibers covered by ZnO NPs (effects at 460 °C and 530 °C). The thermal analysis indicated that GNF@ZnO 2 h and GNF@ZnO 5 h composites included a similar amount of ZnO NPs (~40 wt.%), whereas soaking the gelatin nanofibers in solutions of forming ZnO NPs for 24 h allowed composites with a much higher amount of attached ZnO NPs to be obtained (Table 2).

#### 3.2.4. Photoluminescence Properties of GNF@ZnO Composites

Figure 4 shows normalized PL spectra for pristine ZnO, GNF, and GNF@ZnO composites obtained at different synthesis times. Under excitation, 325 nm pristine ZnO NPs demonstrated two peaks centered at 380 nm and 620 nm. The PL peak in the ultraviolet UV region (380 nm) corresponded with the near band edge emission (NBE) [29,30]. The broad peak in the visible range was associated with defective PL, so-called deep level defects (DLE) [29,31]. It is considered that zinc and ionized oxygen vacancies, as well as surface defects, determine PL intensity in the visible range [32]. They were seen as two peaks at 445 nm and 540 nm in the spectrum of gelatin nanofibers. It should be noted that this value of PL corresponds with the PL obtained by Azhniuk et al. and Li et al. [33,34].

After the synthesis of GNF@ZnO composites, the intensity of PL became 50% higher than for pristine ZnO. The most intense PL was observed for the GNF@ZnO 5 h sample. PL spectra of GNF@ZnO revealed that the visible PL peak was shifted down to 545 nm compared with the pristine ZnO (620 nm). It can be assumed that surface defects, which determine the PL in the orange-red region, had been passivated during GNF@ZnO composites synthesis.

#### 3.2.5. FTIR Analysis of Studied Samples

FTIR analysis is an effective method to reveal the composition of products. The IR spectrum of ZnO NPs consisted of characteristic peaks for ZnO, with main peaks at 545 cm^−1^ and 680 cm^−1^ attributed to the Zn–O bond (Figure 5a) [35]. The spectrum in Figure 5b corresponds with typical pristine gelatin fibers, with C=O stretching vibration appearing at 1664 cm^−1^, demonstrated by the amide I band, and the N–H bending vibration at 1527 cm^−1^, corresponded to the amide band II [36]. Additionally, aliphatic C–H bending vibrations were observed at 1450 cm^−1^, and small bands at 1331 and 1230 cm^−1^ showed the C–N bond stretching vibrations.

The spectra of the GNF@ZnO composites consisted of signals derived from both ZnO NPs and gelatin originating bonds (Figure 5c–e). The peaks at 545 cm^−1^ in composites increased with the increasing ZnO content and indicated the effective adsorption of ZnO NPs on the GNF surface. The broad bands in the 3375–3000 cm^−1^ range for all samples studied were assigned to the O–H stretching mode of surface hydroxyl groups.

#### 3.2.6. Antibacterial Activity

As was mentioned above, antibacterial activity depends on material type, particle size, concentration, and the cell wall structure of the bacteria [37]. Thus, due to its unique characteristics, ZnO is an ideal candidate to study for its antibacterial activity. To evaluate the antibacterial activity of the synthesized pristine ZnO NPs and GNF@ZnO composites, four bacterial strains were selected belonging to Gram-positive (*Staphyloccocus aureus* and *Bacillus pumilus*) and Gram-negative (*Escherichia coli* and *Pseudomonas fluorescens)* strains as model bacteria. The antimicrobial properties were evaluated by disk diffusion assay, as it has been used not only in research concerning ZnO NPs but also with distinct types of nanoparticles, such as copper and silver nanoparticles [23,38,39]. The inhibition zone diameters of the samples against bacteria were shown in Table 3 and Figure 6. The strains most sensitive to GNF@ZnO (2 and 5 h) were *P. fluorescence*, then *S. aureus*, *E. coli*, and *B. pumilus*. It is worth noting that *B. pumilus* were sensitive only for ZnO NPs, whereas the composites with gelatin nanofibers did not retain antibacterial properties against these bacteria (Figure 6, Table 3).

In all cases, the pristine ZnO NPs exhibited the highest antibacterial properties, whereas the gelatin nanofibers were fully biocompatible and showed no antibacterial activity. The high antibacterial activity of pristine ZnO NPs was caused by the small particles size (~7 nm), and consequently, high surface area which led to effective penetration of ZnO NPs into the bacteria membranes.

In the case of the *E. coli* strain, the shortest time of gelatin nanofibers incubation with ZnO seemed to be the most effective, indicating the highest antibacterial properties of this composite. On the other hand, in *S. aureus*, two types of gelatin fibers composites were the most effective (2 h and 24 h). In the case of *P. fluorescens*, the results for all types of gelatin fibers composites were comparable, and surprisingly, the largest zone of growth inhibition was found for GNF@ZnO 5 h. Moreover, the composites also limited the growth of bacteria without completed inhibition (Figure 6c). Interestingly, there is no correlation between sensitivity to composites and bacteria type (Gram-negative vs. Gram-positive). However, in the case of the Gram-positive *B. pumilus* strain, only pristine ZnO effectively inhibited their growth. It is known that the cell wall of Gram-positive bacteria is thick and composed of multilayer peptidoglycan, which could hamper the internalization of nanoparticles. Moreover, the bacterial cell wall of *B. pumilus*, due to specific composition [40], is able to reduce metal toxicity [41] and accumulate the Zn^2+^ ions, which was presented by Ramstedt et al. [42].

The mechanisms of the antibacterial activity can be related to the release of Zn^2+^ ions, which were penetrating and then killing bacteria, as well as to oxidative stress due to the production of reactive oxygen species (ROS), which form oxidizing and highly reactive radicals (O^2−^**•**). Due to ROS, the bacteria can be effectively destroyed or at least deactivated by a reaction of the radicals and the macromolecules, such as DNA, enzymes (protein), lipids, etc. [43]. Thus, the low sensitivity GNF@ZnO composites against *B. pumilus* can be related to barrier properties against ROS and indicated no release of the Zn^2+^ into the bacteria, probably due to large thickness [37].

Since ZnO has excellent characteristics for bioapplication, a large number of publications describe the antibacterial activity of ZnO with different morphologies and their composites. For example, Thanusha et al. obtained hydrogel composite based on gelatin and bioactive components, including ZnO NPs [44]. Antibacterial activity was studied using *E. coli* and *S. aureus.* Our results using ZnO NPs and GNF have higher antibacterial activity in similar bacteria, and a little less than gelatin-based nanocomposite containing chitosan nanofiber and ZnO nanoparticles obtained in [37]. To compare the antibacterial activity of previously reported results with our studies, the respective data sets are presented in Table 3. The composites based on 4A zeolite mixed with semiconductor oxides (TiO_2_, ZnO, and their mixture, TiO_2_/ZnO) were obtained using wet chemistry methods, and their antimicrobial as well as antibacterial activities were studied against four types of bacteria (*E. coli*, *Listeria monocytogenes*, *P. fluorescens*, and *S. aureus*) for the first time [45]. The results showed that the most sensitive bacteria to the ZnO/4A composite were *P. fluorescens* and *E. coli.* Another group studied the antibacterial assay against *S. aureus* and *E. coli* bacterial strains using pristine ZnO NPs with a size of approximately 50 nm [46]. The obtained results showed that the zone of inhibition of the ZnO NPs on *S. aureus* and *E. coli* was 6 mm. Thus, from our results, we can conclude that the small ZnO NPs that covered the GNF are potential agents against the various types of bacteria.

## 4. Conclusions

ZnO, gelatin nanofibers, and GNF@ZnO composites were successfully synthesized through facile and low-cost methods. Their structural, morphological, and luminescence properties were studied. Their antimicrobial activity toward Gram-positive (*S. aureus, B. pumilus*) and Gram-negative (*E. coli*, *P. fluorescens*) bacteria were performed. Results show that the mean size of as-obtained ZnO nanoparticles was approximately 7 nm. Particles were highly crystalline and monodispersed, with a high specific surface area of 83 m^2^/g. The thickness of the gelatin fiber obtained by the electrospinning technique was around 1 µm. SEM, EDX, as well as FTIR analyses confirmed the adsorption of ZnO NPs on the gelatin fibers surface. The antibacterial properties of the obtained composites were investigated by disk diffusion assay on agar plates. The pristine gelatin nanofibers did not show any antibacterial activity for any of the selected bacteria types. At the same time, pristine ZnO NPs exhibited the highest antibacterial properties, caused by the high surface area of the ZnO NPs. In the case of the GNF@ZnO composites, results did not show a correlation between the sensitivity of the composites and bacteria type. The most pronounced sensitivity to GNF@ZnO (2 h and 5 h) was observed for *P. fluorescence* strains. Hence, the proposed composites, prepared by simple approaches and due to their obtained characteristics, can have a potential application in the food industry as antibacterial inhibitor agents.

## Figures and Tables

**Figure 1 materials-14-00103-f001:**
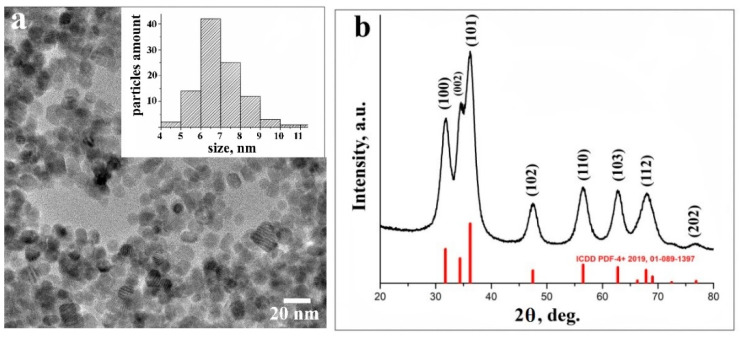
High-resolution transmission electron microscopy (HRTEM) image of ZnO nanoparticles (NPs), with particles size distribution (inset) (**a**); XRD data of ZnO NPs (**b**).

**Figure 2 materials-14-00103-f002:**
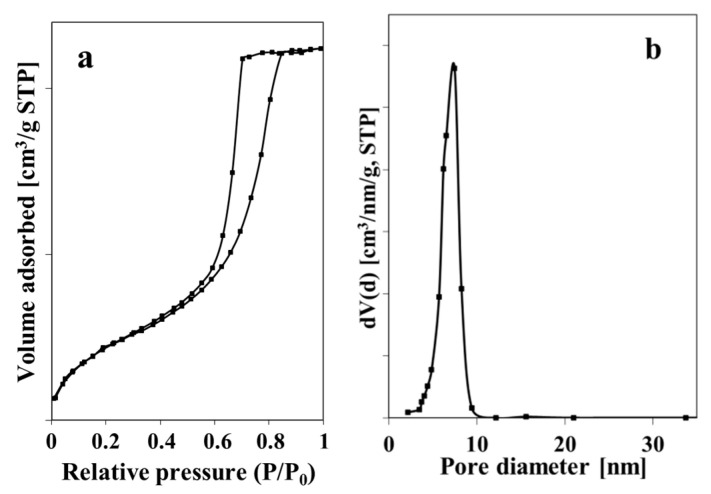
N_2_ adsorption–desorption isotherm (**a**) and particles size distribution (**b**) of ZnO NPs.

**Figure 3 materials-14-00103-f003:**
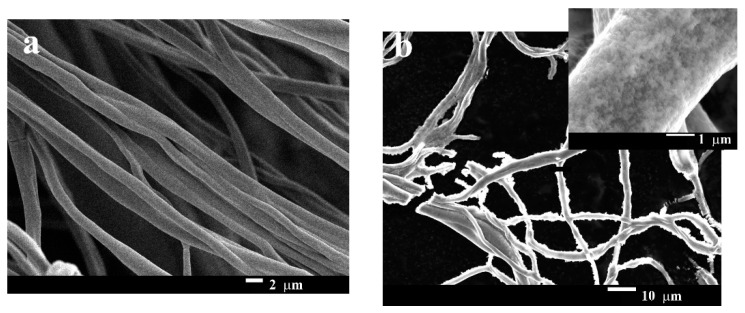
SEM images of pristine gelatin nanofibers (**a**) and gelatin nanofibers covered by ZnO NPs (GNF@ZnO) (**b**).

**Figure 4 materials-14-00103-f004:**
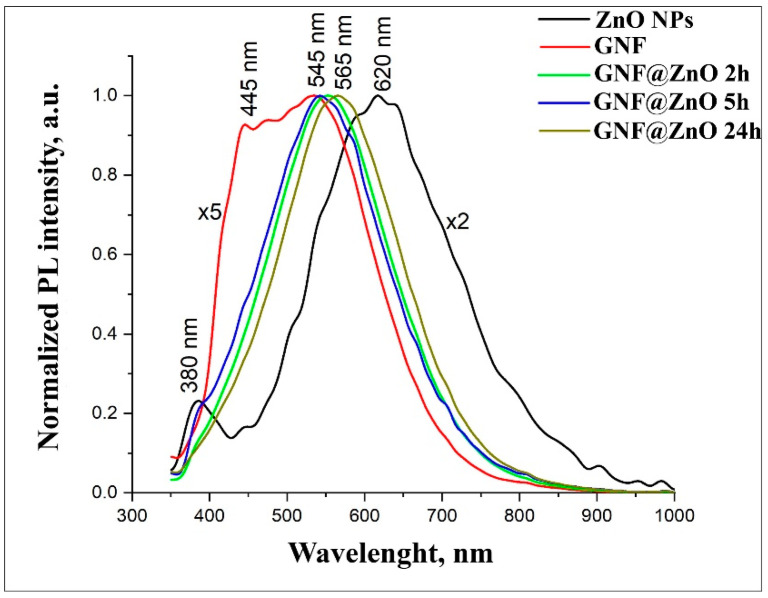
Normalized PL spectra of ZnO, gelatin nanofibers (GNF), and GNF@ZnO composites.

**Figure 5 materials-14-00103-f005:**
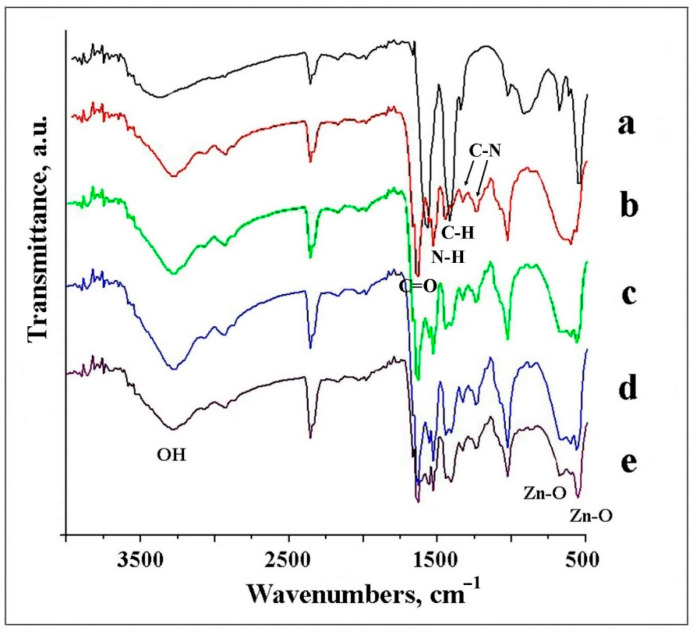
FTIR spectra of the samples: ZnO NPs (**a**), GNF (**b**), GNF@ZnO 2 h (**c**), GNF@ZnO 5 h (**d**) and GNF@ZnO 24 h (**e**).

**Figure 6 materials-14-00103-f006:**
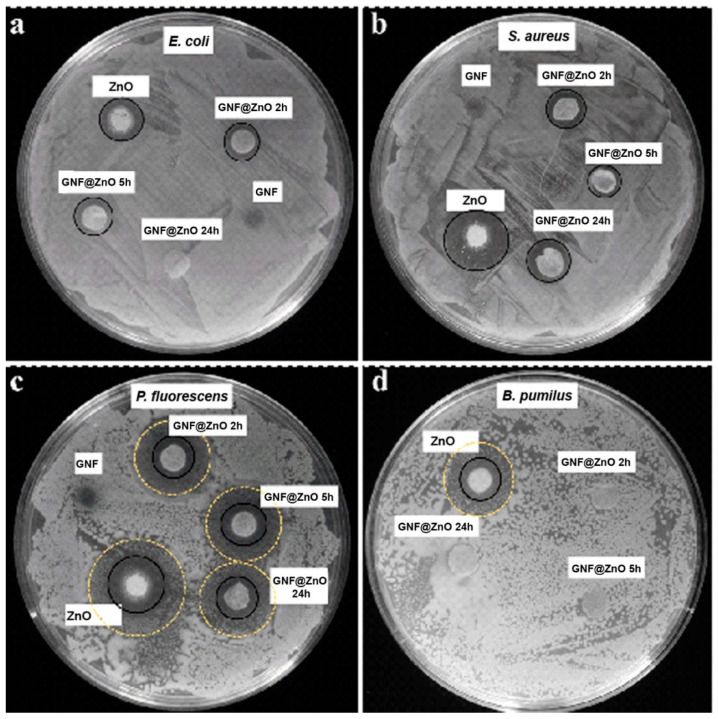
Antibacterial activity of ZnO NPs, GNF, and their composites (GNF@ZnO 2 h, 5 h, 24 h) against *E. coli* (**a**), *S. aureus* (**b**)*, P. fluorescens* (**c**), and *B. pumilus* (**d**) bacteria strains performed on agar plates. Black circles indicate growth inhibition zones, and yellow dashed circles indicate restricted/limited growth zones.

**Table 1 materials-14-00103-t001:** Elemental composition of the studied samples.

	Composite	GNF@ZnO 2 h, Atomic, %	GNF@ZnO 5 h, Atomic, %	GNF@ZnO 24 h, Atomic, %
Elements				
**C**		53.01	50.04	27.11
**O**		31.03	33.95	32.34
**Zn**		15.87	16.01	40.55
**total**		100.00	100.00	100.00

**Table 2 materials-14-00103-t002:** The thermal analysis data.

Sample	1st Weight Loss, %	2nd and 3th WEIGHT Loss, %	Content of ZnO ^a^, wt.%	Content of ZnO ^b^, wt.%
GNF@ZnO 2 h	7.1	55.1	37.8	40.7
GNF@ZnO 5 h	7.0	51.7	41.3	44.4
GNF@ZnO 24 h	4.4	32.2	63.4	66.3

Notes: ^a^ ZnO content in samples contained moisture, ^b^ ZnO content recalculated on dry sample.

**Table 3 materials-14-00103-t003:** Mean diameter of the zone of inhibition (in mm, including the 6 mm diameter of the disk) after ZnO NPs, GNF, and GNF@ZnO composites treatment.

Samples	*S. aureus (G+)*	*B. pumilus (G+)*	*E. coli (G−)*	*P. fluorescens (G−)*
ZnO NPs ^x^	19 ± 0.82	16 ± 0.32	11.3 ± 0.47	17.7 ± 0.47
GNF@ZnO 2h ^x^	10 ± 0.63	0	9.2 ± 0.75	10 ± 0
GNF@ZnO 5h ^x^	7.8 ± 0.4	0	8.9 ± 0.49	11.1 ± 0.8
GNF@ZnO 24h ^x^	11 ± 0	0	0	9.8 ± 0.75
GNF ^x^	0	0	0	0
gelatin + ZnO *	4.9 ± 0.6		5.3 ± 0.2	
G.ZnO NP ** G.CHNF.ZnO NPs **	30.62 ± 0.56 33.13 ± 0.67		15.06 ± 0.17 25.06 ± 0.24	
ZnO/4A z ***	6.21 ± 0.02		6.86 ± 0.03	6.34 ± 0.03
ZnO NPs (~50 nm) ****	6.0		6.0	

**Notes**: ^x^—our results; *—[44]; **—[37]; ***—[45]; ****—[46].

## Data Availability

The data presented in this study are available on request from the corresponding author.

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
