# Peer review of "Fabrication of Gelatin-ZnO Nanofibers for Antibacterial Applications"

_materials, 2020, doi:10.3390/ma14010103_

Round 1
Reviewer 1 Report
The submitted manuscript is about physico-chemical characterization and antibacterial characteristics of GF@ZnO composites, some interesting results were obtained. However, everything was spoiled with bad language, generally background writing and low-level language errors. Some Points are list below:
- The obtained materials (GF@ZnO nanoparticles in title; GF@ZnO NPs nanocomposites in Abstract) , should be corrected, GF@ZnO composites, and the abbreviations used commonly in the manuscript.
- Generally, entire Introduction chapter looks as a patchwork of words in sentences and sentences in paragraphs, sometimes with some sense, sometimes without, but always annoying. Partial statements can be deleted, for example, Introduction (line 65-70), and the paragraph needs simplification.
- ZnO NP synthesis (line 99), there are many miss written words in language expression, E.g. described in [9], Zn((CH3COO)2×2H2O), was added to zinc acetate dihydrate? After about 1.5h white powder was started to precipitate indicating the formation of ZnO NPs.? The experiment phenomenon should be deleted in this part. Especially, the preparation of GF@ZnO composites should be listed in this part, and the synthesis of ZnO NPs takes second place, and antibacterial activity of the obtained composites should be corrected into antibacterial activity of GF@ZnO composites.
- In part 3.2.1, Figure 2(b), inset, the high resolution images is needed in this part though the rough surface occurs, e.g. magnification ×2000, ×4000, ×7000? the distribution of ZnO nanoparticles?, low-level language errors in Figure 3.
- The data "surface area of 83 m2/g"? please repeat measurement.
- The mechanisms of the antibacterial activity can be related with release of Zn2+?Please provide the data of Zn2+ ion precipitation.
Author Response
We thank you and the Reviewer for taking the time to evaluate our manuscript, for the comments, questions, criticism, and suggestions.
Please find an extensive list of all replies to the Reviews below. We believe we were able to revise and improve our manuscripts in light of all the comments and that it merits publication in Materials in its current form.
All comments were taken into account, the corrections and additional information are included in the manuscript and supporting information. All changes were marked in red.

Reviewer 2 Report
In the present article, the author has shown wet-chemical synthesis of gelatin-ZnO nanoparticles can be used as antibacterial agents. The work is interesting however, there are some errors and experimental lacking which have to rectify for further consideration. I, therefore, suggest the paper can be considered subject to major changes.
- The Title is can be changed to “Fabrication of gelatin-ZnO nanofibers for antibacterial applications”. The title is changed according to the author's research work.
- In Abstract GF should be replaced with “GNF” (gelatin nanofibers).
- In Abstract Zn2+ should be written as “Zn2+”.
- The whole manuscript contains lots of grammar errors also the author used more complex sentences. I suggest the author should use some editorial service or native speaker for language editing or Grammarly, Inc. software.
- The abstract should be improved by adding result values especially for TEM and antibacterial results with the zone of inhibitions.
- In the Introduction the author showed that “Nourbakhsh et al. obtained composites based on gelatin/ZnO nanocomposite films and studied their antibacterial properties [25]” Then what is the improvement in this research should be elaborated in the introduction section after reference [25].
- In Introduction line 77 and 78 authors should be specific in their synthesis procedure as sol-gel and the wet chemical is different. In Sol-gel synthesis the precursors are converted into a gel(sol) whereas in the wet-chemical process the precursors are converted into precipitate.
- GF@ZnO composite section should be described in detail as it is the main content of this research.
- Inline number 128 to 129, always the bacteria name should be written in the italic letter.
- The disc diffusion method should be cited with some references.
- In line 130 the 106 CFU/ml should be “106 CFU/ml” please change it.
- In Figure 1b. XRD should be added with ICDD PDF files cited database pattern.
- The BET results graphs should be presented in the paper.
- The SEM images with 2h, 5h, and 24 h should be presented with elemental mapping to show the attachment of ZnO in it.
- The mechanism of photoluminescence properties of GF@ZnO composites should be presented.
- TG/DTA results are also required to prove the amount of ZnO attached to the GF@ZnO composites
- In Figure 5a why there is no zone of inhibition in GF@ZnO 24h samples explain the reason, also why no zone of inhibition in B. pumilus explain the reason in the results and discussion section.
Author Response
We thank you and the Reviewers for taking the time to evaluate our manuscript, for the comments, questions, criticism, and suggestions.
Please find an extensive list of all replies to the Reviews below. We believe we were able to revise and improve our manuscripts in light of all the comments and that it merits publication in Materials in its current form.
All comments were taken into account, the corrections and additional information are included in the manuscript and supporting information. All changes were marked in red.
"Please see the attachment."

Round 2
Reviewer 1 Report
I have read the responses to the referees’ comments and the revised version of manuscript. The suggestions were inserted in the text. The quality manuscript was improved. Now, my suggestion is that this manuscript can be accepted to be published in this Journal.
Reviewer 2 Report
The authors have responded to the queries well, I recommend the paper can be accepted for publication.